# Car Price Quotes Driven by Data-Comprehensive Predictions Grounded in Deep Learning Techniques

Andreea Dutulescu [1,2], Andy Catruna [1,2], Stefan Ruseti [1], Denis Iorga [3], Vladimir Ghita [2,4], Laurentiu-Marian Neagu [1,2] and Mihai Dascalu [1,2,*]

1   Computer Science Department, University Politehnica of Bucharest, 060042 Bucharest, Romania; andreea.dutulescu@stud.acs.upb.ro (A.D.); andy_eduard.catruna@stud.acs.upb.ro (A.C.); stefan.ruseti@upb.ro (S.R.); laurentiu.neagu@upb.ro (L.-M.N.)

2   R&D Department, Global Resolution Experts, 061344 Bucharest, Romania; vladimir.ghita@grx.ro

3   Interdisciplinary School of Doctoral Studies, University of Bucharest, 030018 Bucharest, Romania; denis.iorga@drd.unibuc.ro

4   Management Department, University Politehnica of Bucharest, 060042 Bucharest, Romania

*   Correspondence: mihai.dascalu@upb.ro

**Abstract:** The used car market has a high global economic importance, with more than 35 million cars sold yearly. Accurately predicting prices is a crucial task for both buyers and sellers to facilitate informed decisions in terms of opportunities or potential problems. Although various machine learning techniques have been applied to create robust prediction models, a comprehensive approach has yet to be studied. This research introduced two datasets from different markets, one with over 300,000 entries from Germany to serve as a training basis for deep prediction models and a second dataset from Romania containing more than 15,000 car quotes used mainly to observe local traits. As such, we included extensive cross-market analyses by comparing the emerging Romanian market versus one of the world's largest and most developed car markets, Germany. Our study used several neural network architectures that captured complex relationships between car model features, individual add-ons, and visual features to predict used car prices accurately. Our models achieved a high $R^2$ score exceeding 0.95 on both datasets, indicating their effectiveness in estimating used car prices. Moreover, we experimented with advanced convolutional architectures to predict car prices based solely on visual features extracted from car images. This approach exhibited transfer-learning capabilities, leading to improved prediction accuracy, especially since the Romanian training dataset was limited. Our experiments highlighted the most important factors influencing the price, while our findings have practical implications for buyers and sellers in assessing the value of vehicles. At the same time, the insights gained from this study enable informed decision making and provide valuable guidance in the used car market.

**Keywords:** car price prediction; visual features; cross-market analysis; feature analysis; deep neural networks

## 1. Introduction

### 1.1. Overview

The used car market plays an important role in the global automotive industry, offering an alternative avenue for consumers to purchase vehicles at a lower price than new cars. Moore [1] argued that more than 35 million cars are sold yearly. This market encompasses buying and selling pre-owned vehicles typically obtained through trade-ins, auctions, or private sales. The used car market dynamic varies across different countries due to variations in consumer preferences, economic factors, and regulatory frameworks. Understanding these variations and predicting used car prices is essential for buyers and sellers to make informed decisions and negotiate fair transactions.

The used car market has witnessed substantial growth and transformation in various countries over the years. While developed economies like Germany have long-established used car markets, emerging economies have experienced rapid expansion in recent decades. This growth can be attributed to increasing disposable incomes, growing urbanization, and shifting consumer preferences toward affordable transportation options. Furthermore, differences in government policies, taxation, and import regulations have shaped the unique characteristics of each country's used car market.

A comprehensive study of car markets, including new and used segments, is crucial as it could provide insights into the overall health and dynamics of the automotive industry, which serves as a key economic indicator. Applying machine learning techniques to study used car markets has gained high levels of attention recently. Machine learning algorithms offer a powerful means to analyze vast amounts of data and extract meaningful patterns, thereby enabling the prediction of car prices based on their inherent features. The objective of employing machine learning in this context is to develop accurate and reliable models to estimate the value of a used car based on its make, model, year of manufacture, mileage, condition, and other relevant attributes. In doing so, machine learning techniques facilitate informed decision making for buyers and sellers, enabling them to assess the fair market value of a vehicle and negotiate prices more effectively.

Conducting a comprehensive and comparative study on two distinct car markets would also involve exploring the efficacy and generalizability of machine learning techniques for used car price prediction. This comparative approach would support identifying similarities, differences, and factors contributing to variations in used car prices across different markets. By selecting two countries with diverse economic, cultural, and regulatory contexts, this study aimed to highlight the contextual factors that influence the pricing practices of used cars. Moreover, we sought to evaluate the performance of machine learning models in predicting prices in these markets, thereby contributing to the development of robust prediction methodologies to be applied across various car markets worldwide.

### 1.2. Related Work

### 1.2.1. Estimating Used Car Prices

Estimating used car prices is a significant research area that poses a challenge for scholars aiming to address the regression task associated with price prediction. We emphasize that no universally recognized solution exists, and existing studies predominantly focus on specific localized markets. Both emerging economies and developed countries boast substantial second-hand car markets, and research endeavors in this domain span diverse regions, including Germany [2], Bosnia and Herzegovina [3], China [4,5], India [6], Bangladesh [7], and Romania [8].

For this task, we chose studies that have employed diverse approaches across various market locations. Some investigations have used simple machine learning algorithms, such as linear regression, decision trees [9], and gradient boosting [10]. These algorithms are favored due to their interpretability and fast convergence for small- to medium-scale datasets. In contrast, neural networks have been employed to increase prediction accuracy by improving the aggregation of complex information extracted from features of different kinds. The next subsections describe, in chronological order, six approaches that employed simple regressors, followed by two studies that experimented with classical neural networks, while the last two papers tackled how deep neural network architectures like CNNs and transformers can fuse information in the prediction process.

Classic Machine Learning

Pal et al. [2] developed a model for car price prediction using a random forest classifier [11]. Their dataset comprised 370,000 German eBay entries related to the prices and attributes of used cars. The data preprocessing and exploration procedure resulted in using only 10 out of the 20 car attributes from the initial dataset, namely: car shape, brand, model, age, mileage, engine power, type of fuel, transmission, whether the car was

damaged and repaired or not, and price. Following model training and testing, the authors obtained an $R^2$ of 0.83 on the validation data, with price, kilometers, brand, and vehicle type being the most relevant features.

Kondeti et al. [7] implemented various machine learning techniques to develop a model for car price estimation. The dataset used by the authors consisted of 1209 entries related to the prices and attributes of pre-owned cars. It was obtained via scraping methods applied on an online marketplace from Bangladesh. Here, again, data were explored and preprocessed to address issues related to outliers, missing data, unrepresentative samples, the lack of numerical representations of text attributes, multicollinearity, and different measurement scales. This resulted in using 9 of the 10 car attributes present in the initial dataset, namely transmission, fuel type, brand, car model, model year, car shape, engine capacity, mileage, and price. Of the five implemented regression models, extreme gradient boosting was declared most suitable for car price prediction, with an $R^2$ of 0.91, closely followed by the random forest classifier with an $R^2$ of 0.90. However, random forests scored better in terms of the mean average error.

Gegic et al. [3] created a car price prediction model based on an ensemble architecture consisting of a random forest classifier, a neural network, and a support vector machine [12]. They collected a dataset for used car price estimation in the market of Bosnia and Herzegovina by utilizing a web scraper and cleaning up the data, resulting in 797 distinct samples. The inputs of the models were features such as the brand, model, car condition, transmission, mileage, and color. The authors initially converted distinct price intervals to nominal classes and tested each type of classifier separately, obtaining subpar performance. Subsequently, they introduced an intermediate task of classifying the cars into "cheap", "moderate", or "expensive", which was performed by the random forest classifier. Based on this result, the features were given as the input for an independent support vector machine or neural network, further refining the prediction by estimating the class of the price interval. Their final architecture combining all the classifiers obtained an accuracy of 0.87.

Venkatasubbu and Ganesh [13] experimented with different supervised regression techniques for used car price prediction and studied which variables were most predictive for this task. They considered the dataset introduced by Kuiper [14], which contained a total of 804 sample cars with annotations for mileage, make, model, trim, body type, cylinder, liters, doors, cruise, sound, leather seats, and price. The authors trained models for lasso regression [15], multiple linear regression, and regression trees on a training set consisting of 563 records, leaving the rest of the samples for testing. The multiple regression model obtained the lowest error rate of 3.468%, the regression tree obtained an error rate of 3.512%, and the lasso regressor obtained an error rate of 3.581%.

Samruddhi and Kumar [6] tackled the task of car price prediction with a k-nearest neighbor classifier. Their experiments were conducted on a Kaggle dataset containing information about each car's name, location, year, kilometers, fuel, transmission, mileage, owner's number, engine, power, and seats. They encoded these values to obtain a high-dimensional Euclidean space in which they utilized a k-nearest neighbor algorithm to predict the price. The price of an unknown sample was predicted as the average of the closest $k$ known cars in this Euclidean space. They performed an analysis to obtain the optimal $k$ value, and their results showed that observing the closest four samples yielded the lowest error rate. Their model obtained an accuracy of 82%, a root mean square error (RMSE) rate of 4.74, and a mean absolute error (MAE) rate of 2.13.

Gajera et al. [16] used a dataset consisting of 92,386 records to train multiple regression techniques such as KNN regression, random forest, linear regression, decision trees, and XGBoost. Each sample contained information about mileage, the year of registration, fuel type, car make, model, and gear type. The random forest regression model obtained the lowest error rate and achieved an RMSE of 3702.34, followed by the XGBoost model, which obtained an RMSE of 3980.77.

Neural Networks—Multi-Layer Perceptron

Liu et al. [5] proposed a PSO-GRA-BPNN (particle swarm optimization–grey relation analysis–backpropagation neural network) model for second-hand car price prediction. The dataset collected for the implementation of the model comprised 10,260 entries related to the attributes of second-hand cars sold through a car trading platform from East China. The attributes used for developing the model were: brand, drive mode, gearbox, engine power, car shape, mileage, age, fuel consumption, emission standard, region, and price. The performance of the PSO-GRA-BPNN model was compared to random forest, multiple linear regression, and support vector machine models. Based on this comparison, the authors concluded that the performance of the PSO-GRA-BPNN model was superior to that of the others, with an $R^2$ of 0.98 and a mean average percentage error of only 3.9%. However, their model was the slowest in terms of training speed compared to the other models.

Cui et al. [4] introduced an innovative framework for price regression, employing a combination of two gradient-boosting techniques and a deep residual network [17]. The authors conducted experiments on a dataset comprising more than 30,000 samples, taking into account over 20 features, including the most frequently used features like the car brand, mileage, age, and fuel type. The neural network processed the input features, generating an optimized representation of the attribute characteristics. This representation and the initial prediction served as the input for an XGBoost module, which iteratively predicted the price by incorporating the predicted price from the previous iteration and the initial features. To further enhance the results, a LightGBM framework was employed, utilizing the preceding prediction and initial features to retrain the representations iteratively until performance improvement plateaued. The proposed evaluation metric, which combined the mean absolute percentage error (MAPE) and accuracy, yielded a score of 75 out of 100.

Deep Neural Networks

Yang et al. [18] studied the problem of car price prediction from images by employing multiple classic machine learning techniques and deep learning models such as convolutional neural networks. They constructed a dataset consisting of 1400 images of front angular views of different cars, with prices ranging from USD 12,000 to USD 2,000,000. They developed initial baselines for price regression based on linear regression models that took as their input HOG features or features extracted from pre-trained CNNs. Moreover, they created a classification task by splitting the data into price intervals and training the models to predict the price class. The researchers allocated class segments to each example by employing price cutoffs that aligned with specific percentiles of price distribution (20th, 40th, 60th, 80th, and 100th percentiles) to predict car prices. Their baseline consisted of a support vector machine classifier for this task. They further analyzed the performance of CNN models such as SqueezeNet [19] and VGG-16 [20], along with a custom architecture, PriceNet, which built upon SqueezeNet by adding residual connections between modules and batch normalization. The PriceNet architecture achieved the best performance for all metrics, obtaining an RMSE of 11,587.05, an MAE of 5051.61, an $R^2$ score of 0.98 for the regression task, and an F1 score of 0.88 for classification.

Dutulescu et al. [8] studied several approaches in terms of price prediction, employing baseline models such as XGBoost and experimenting with deep neural networks to better aggregate car features. They constructed a dataset of 25,000 ads from a Romanian website that advertised used cars. The features used in the prediction were the brand; model; year of manufacture; mileage; fuel; engine capacity; transmission; and a list of add-ons, which were extra components of cars that customers could opt to include on their cars. The employed neural networks learned embeddings for the car model to better represent this feature, and several experiments were performed for add-on representation. Add-ons were represented as their total count, hot-encoded with a dense projection, or encoded as trainable embeddings with and without a self-attention layer. Moreover, a pre-trained RoBERTa model was employed on the text descriptions of the add-ons to capture the

linguistic meaning of these options. The best scores of 95.47 $R^2$ and 10.68% mean percentage error were obtained by the neural network that employed learned embeddings on add-ons.

The problem with most of the identified studies was their small scale in terms of dataset size. However, deep neural networks, which exceed the performance of simple models in every task nowadays, require a large training set for capturing the complex relations between the features to be successfully employed. The current landscape of car price prediction would benefit from deep learning approaches that can take full advantage of car feature information. Moreover, a comprehensive study of multiple markets and their particularities has yet to be conducted, as the prediction models' potential is far from being fully explored.

### 1.2.2. Computer Vision Models for Image Processing

In terms of image analysis architectures, SqueezeNet [19] is a deep neural network specifically designed for efficient image classification tasks. It stands out due to its model compression technique, achieving high accuracy while reducing the model size and computational complexity. The key innovation of SqueezeNet lies in its fire module, which combines both squeeze and expand operations to strike a balance between model efficiency and expressive power. SqueezeNet has demonstrates good performance on various benchmark datasets. It has achieved comparable or superior results to deeper and larger networks while having considerably fewer parameters.

EfficientNet Tan and Le [21] is a newer convolutional neural network that has achieved superior performance by balancing model complexity and computational efficiency. The architecture employs a compound scaling technique that uniformly scales the network's depth, width, and resolution, improving accuracy while minimizing computational overhead. EfficientNet has consistently achieved top performance in well-known challenges, such as ImageNet [22] classification (84.3% accuracy), and outperformed previous models [17,23,24] by a large margin. Moreover, EfficientNet has demonstrated its effectiveness in transfer-learning scenarios, where it excels at learning representations from large-scale pre-training datasets and transferring that knowledge to downstream tasks with limited labeled data.

Liu et al. [25] built upon the vision transformer (ViT) architecture [26], which used the self-attention mechanism introduced in the transformer model [27] to capture interactions between image patches. Their Swin transformer considered a hierarchical approach to extract features at multiple resolutions, making it suitable as a backbone for multiple vision tasks. This hierarchical structure was obtained by incrementally combining embeddings corresponding to neighboring image patches. Moreover, the Swin transformer replaced the classic self-attention operation requiring a quadratic computation time with a more efficient approximation function. It split the operation into two modules, one that captured the interactions between image patches inside a local window and one that shifted the local window, capturing global information. The Swin transformer obtained state-of-the-art results in image classification on ImageNet [22], object detection on COCO [28], and image segmentation on ADE20K [29].

### 1.3. Research Objective

Our research objective was to develop comprehensive and accurate machine learning models for predicting used car prices in different car markets. As such, this study aimed to train deep learning architectures that captured complex relationships between car features from two distinct datasets localized in Romania and Germany. To this end, we conducted a comprehensive evaluation of three car price prediction approaches based on: gradient boosting as a baseline, neural networks for learning the interactions between the features of the car, and deep image processing architectures for obtaining relevant visual features from the car image. Overall, this research contributes to developing robust prediction methodologies for used car prices to be applied across various car markets worldwide and provides a deeper understanding of the contextual factors that define used car prices,

highlighting the similarities and differences between markets while providing insights into pricing dynamics.

This study expanded upon the initial experiments performed by Dutulescu et al. [8]. Along with improving the initial dataset with additional features, another large-scale dataset was introduced, representative of the German market with over 300,000 car quotes. This served as a resource for our cross-market analysis. We reproduced the experiments on both datasets and proposed new ways of aggregating categorical features. Moreover, we complemented our approach with image analysis to improve the predictions further.

The main contributions of this article toward these objectives are threefold:

- We created a comprehensive dataset comprising over 300,000 entries from the German market and a second dataset comprising more than 15,000 car quotes from the Romanian market. These datasets served as valuable resources for training the deep prediction models and enabled a comparative analysis of the behavior and prediction models between the emerging Romanian and well-established German markets. To our knowledge, these combined datasets represent the largest corpus analyzed to date and provided a basis for the first cross-market analysis on fine-tuning deep learning models.
- We introduced state-of-the-art approaches for developing advanced prediction models that accurately estimated used car prices by considering multiple types of features, trainable embeddings, multi-head self-attention mechanisms, and convolutional neural networks applied to car images. These models achieved a high prediction accuracy, with the $R^2$ score exceeding 0.95. Moreover, we added to these findings an extensive ablation study to showcase the most relevant features, while an error analysis was also performed to study the models' limitations.
- We created a baseline model that employed convolutional architectures to predict car prices based solely on visual features extracted from car images. This model demonstrated transfer-learning capabilities, enabling improved prediction accuracy, particularly for low-resource training datasets. This highlighted the potential for leveraging visual information alone to predict car prices accurately.

## 2. Method

This section describes the introduced datasets and the prediction methods in detail.

### 2.1. Datasets

The datasets used in this study were extracted from two distinct platforms, namely Autovit.ro and Mobile.de, which are prominent websites dedicated to selling second-hand cars in Romania and Germany, respectively. Autovit.ro primarily caters to the Romanian market, with a localized focus limited to the country's geographical boundaries. In contrast, Mobile.de is a broader platform that serves the German second-hand car market, known for having the largest market share within the European Union. However, it is worth noting that Mobile.de is also widely used by individuals in neighboring countries, including Romania. Overall, 30,264 car ads were scraped from Autovit.ro, while 1,308,575 entries were extracted from Mobile.de, on March 2023. As per the terms and conditions statements of each website, posting duplicate ads of the same vehicle is not allowed, and there are methods in place to remove the ads violating this rule. This implies that no duplicate vehicles were part of our dataset. While the same features were extracted from both websites, variations in the feature values were observed, requiring subsequent post-processing steps for normalization. The features considered relevant for the purpose of this investigation encompassed the car brand, car model, year of manufacture, mileage, engine power, gearbox type, fuel type, engine capacity, transmission, car shape, color, add-ons, images, and price.

An outlier filtering technique was employed to ensure the integrity of the data and eliminate spurious ads that could adversely impact the training and prediction processes. This filtering procedure was conducted alongside additional pre-processing steps to maintain data quality.

The subsequent sections detail the pre-processing steps undertaken for both datasets unless otherwise specified.

1.  Mobile.de ads do not explicitly contain the categorized car brand and model but rather a title written by the seller. We extracted these two relevant features from the title using a greedy approach of matching them against an exhaustive list of all car brands and models and choosing the closest fit. Finding a category was impossible for some ads, and these entries were dropped from the dataset.

2.  We discarded the ads that did not contain the relevant features mentioned above and those that did not contain at least an image of the car's exterior. As some sellers published multiple images, some irrelevant to the ad or not showing the entire vehicle, we only considered images that contained the full car exterior. This filtering was carried out with the help of a YOLOv7 model [30] that detected a bounding box for a car image. Images displaying multiple cars without a prominent focus (e.g., parking lots) or with car bounding boxes occupying less than 75% of the entire image size were removed from the dataset.

3.  To maintain precision and minimize the presence of erroneous data, listings with questionable features were eliminated, as they could potentially contain inaccurate information. Thus, we excluded cars with a manufacturing year before 2000, a mileage exceeding 450,000 km, a price surpassing EUR 100,000, and an engine power exceeding 600 horsepower.

4.  The dataset was split randomly into an 80% training set and a 20% validation set; however, we ensured a balanced distribution of car brands in each subset. Notably, a car advertisement could include multiple images, resulting in multiple entries within the dataset (one for each image). However, measures were taken to ensure that the training and validation sets did not contain the same advertisements but rather different ads, each with their respective images.

5.  Within the training dataset, we calculated each car model's mean and standard deviation. Subsequently, we removed outlier listings from the entire dataset that fell outside the range defined by $mean_{model} \pm std_{model}$, where $mean_{model}$ and $std_{model}$ are the mean price and standard deviation calculated for each car model, respectively. When considering car models with less than 20 instances in the dataset, we calculated the mean and standard deviation for the car manufacturer instead to ensure meaningful measurements, because the car manufacturer category contained an adequate number of entries for each group. We determined the mean and standard deviation on a per-model basis for frequently represented car models (i.e., with over 20 sale ads).

This filtering removed 15,253 entries from Autovit.ro and 1,001,324 entries from Mobile.de; thus, our datasets retained 15,011 and 307,251 unique entries, respectively. Moreover, 59,450 images were available for Autovit.ro ads, while 1,628,546 images from Mobile.de were kept.

A thorough analysis of both datasets is presented below. We based our choice of experiments on this analysis to make the most of the data.

The car brand and model were the most relevant categorical features for our task. Figures 1 and 2 depict the frequency distributions of the top 10 most popular car brands in the datasets. Notably, a similarity emerges from the figures, as they reveal a considerable overlap in the most frequently occurring models between the German and Romanian markets. This alignment could be attributed to the substantial influx of car imports into Romania, particularly in the form of second-hand vehicles originating from Germany.

The next features had the same distribution and tended to follow the same pattern from one dataset to another, even though their values were not considered when creating the training and validation partitions. The year of manufacture data in Figures 3 and 4 show that the majority of vehicles were manufactured between 2015 and 2020, with an approximate age of around 5 years at the time of the ad posting.

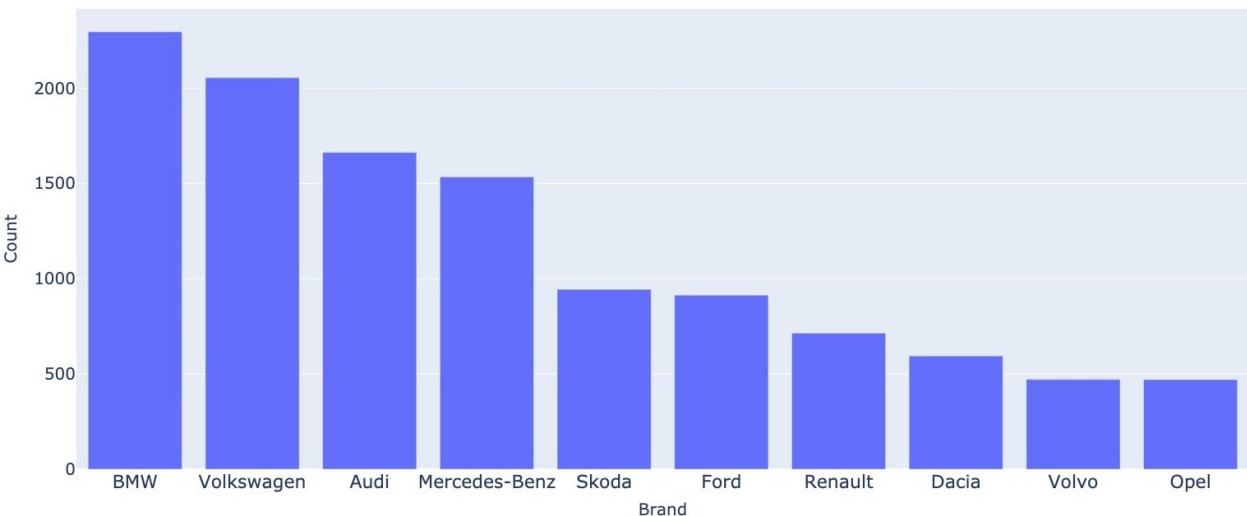

**Figure 1.** Ten most popular brands—Autovit.ro.

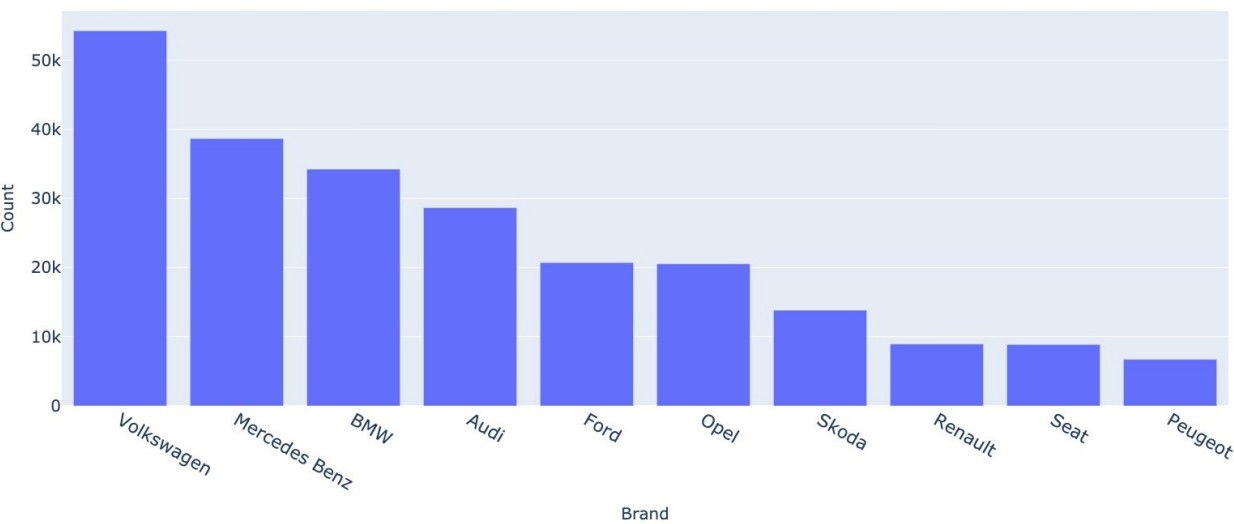

**Figure 2.** Ten most popular brands—Mobile.de.

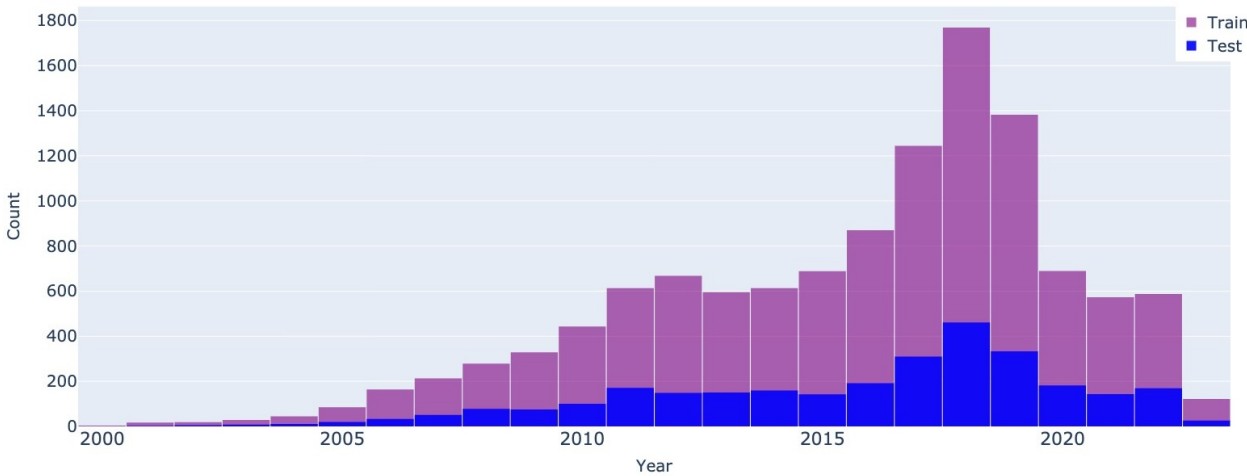

**Figure 3.** Year of manufacture—Autovit.ro.

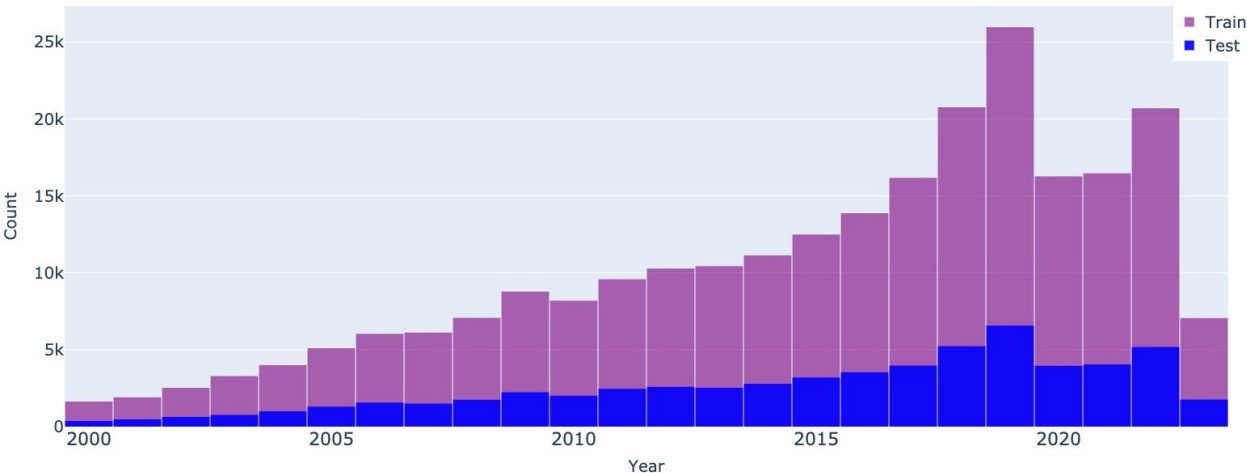

**Figure 4.** Year of manufacture—Mobile.de.

In terms of mileage, roughly 5% of the cars (i.e., 728 from the Romanian dataset and 21,171 from the German ads) had less than 5000 km, making them candidates for new or almost new vehicles. Here, the difference between the two datasets was more striking, as Mobile.de ads tended to become less frequent as the mileage increased. At the same time, a high number of vehicles from the Romanian market were sold at around 200,000 km, a tendency also shown in Figures 5 and 6.

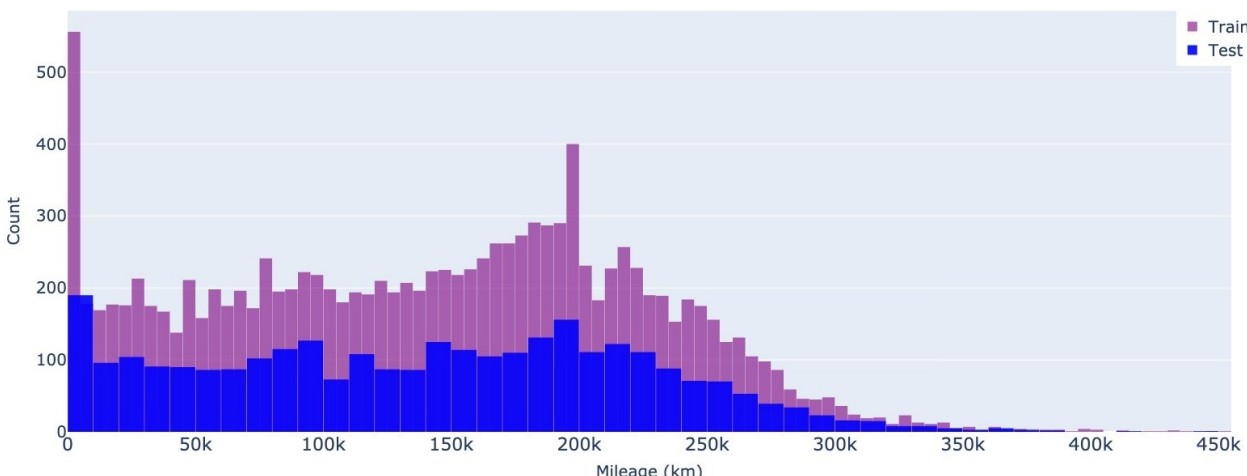

**Figure 5.** Mileage—Autovit.ro.

In terms of engine power, the values were measured in horsepower, and the vast majority of advertised cars had a value between 100 and 200 HP. Off-value ads had a lower frequency for both datasets, especially after the 300 mark. However, Mobile.de also advertises luxury cars with a high engine power given its wider market (see Figures 7 and 8).

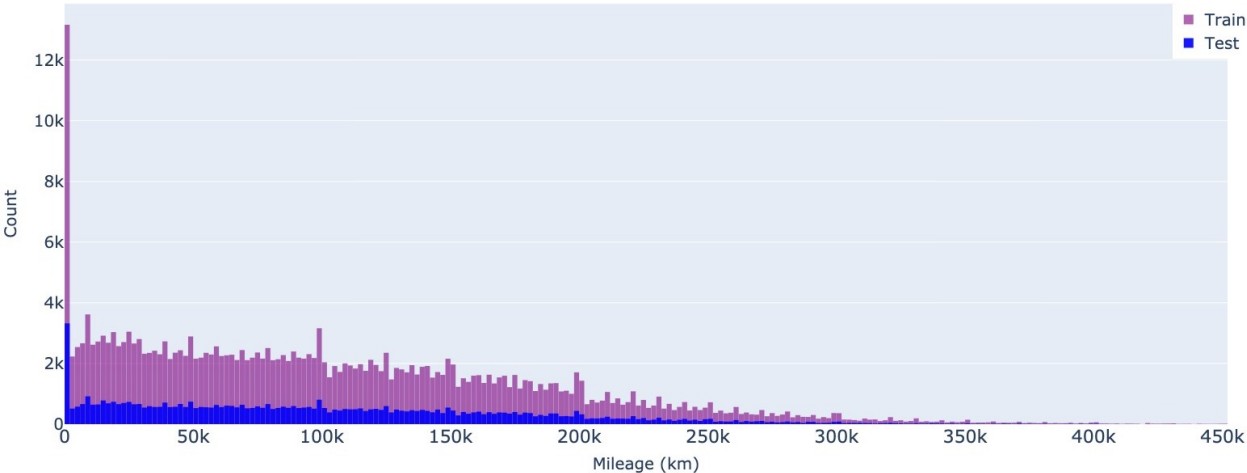

**Figure 6.** Mileage—Mobile.de.

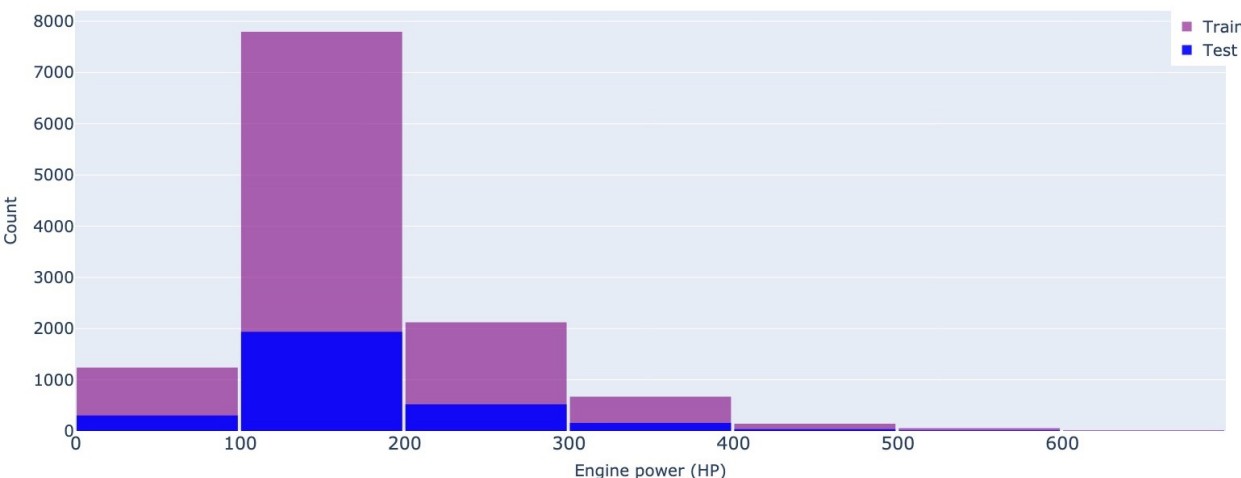

**Figure 7.** Engine power—Autovit.ro.

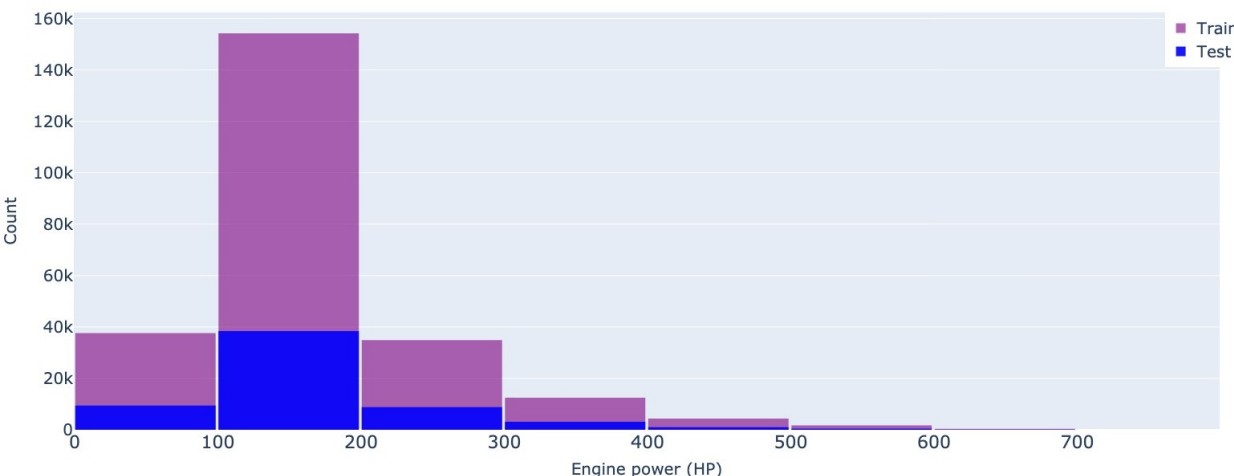

**Figure 8.** Engine power—Mobile.de.

Table 1 highlights the distribution of a subset of features across both datasets and partitions. A difference was observed in the gearbox category between the two datasets. The distribution on Autovit.ro highlighted a strong preference for automatic cars, with around 50% more automatic vehicles than manual ones. The gearbox type tended to have a

meaningful impact on the selling price. However, the difference was small in the German market, and the numbers were balanced between the two classes. The fuel type highlighted another striking difference between the two datasets. Diesel cars were advertised on Autovit.ro more than gasoline-based ones by a large margin. Although this preference was also be observed on Mobile.de, the vehicles were more evenly balanced. However, in both datasets, oil-based fuel was strongly preferred in comparison to alternatives. The engine capacity was a relevant feature, since it influences car tax. It had a similar distribution in both datasets, with most vehicles advertised at around 2000 cm$^3$. In terms of transmission type, a preference for $2 \times 4$ transmission was observed in both datasets, as an integral transmission increased the price; this difference was more pronounced in the Mobile.de dataset. However, it should be noted that while sellers on Autovit.ro were asked to choose the transmission type, on the German website, users had the possibility of adding the $4 \times 4$ feature as an add-on, and many may have omitted this step.

**Table 1.** Distribution of specific features.

| Feature | Value | Autovit.ro | | | Mobile.de | | |
|---|---|---|---|---|---|---|---|
| | | Training | Validation | Total | Training | Validation | Total |
| Transmission | $2 \times 4$ | 6914 | 1690 | 8604 | 192,305 | 47,932 | 240,237 |
| | $4 \times 4$ | 5125 | 1282 | 6407 | 53,561 | 13,453 | 67,014 |
| Fuel | Diesel | 8445 | 2082 | 10,527 | 105,842 | 26,442 | 132,284 |
| | Gasoline | 2893 | 683 | 3576 | 128,643 | 32,172 | 160,815 |
| | Hybrid | 645 | 193 | 838 | 9545 | 2310 | 11,855 |
| | LPG | 56 | 14 | 70 | 1596 | 401 | 1997 |
| | Others | | | | 240 | 60 | 300 |
| Gearbox | Automatic | 7469 | 1876 | 9345 | 127,071 | 31,828 | 158,899 |
| | Manual | 4570 | 1096 | 5666 | 118,795 | 29,557 | 148,352 |
| Engine capacity | <1000 | 508 | 106 | 614 | 22,257 | 5543 | 27,800 |
| | 1000–2000 | 8743 | 2183 | 10,926 | 175,479 | 43,902 | 219,381 |
| | 2000–3000 | 2623 | 642 | 3265 | 39,488 | 9,780 | 49,268 |
| | 3000+ | 165 | 41 | 206 | 8642 | 2160 | 10,802 |

The car shape also had a different distribution among the two datasets, as SUVs are prevalent in the Romanian market. At the same time, the Mobile.de website presented a more balanced distribution (see Figures 9 and 10).

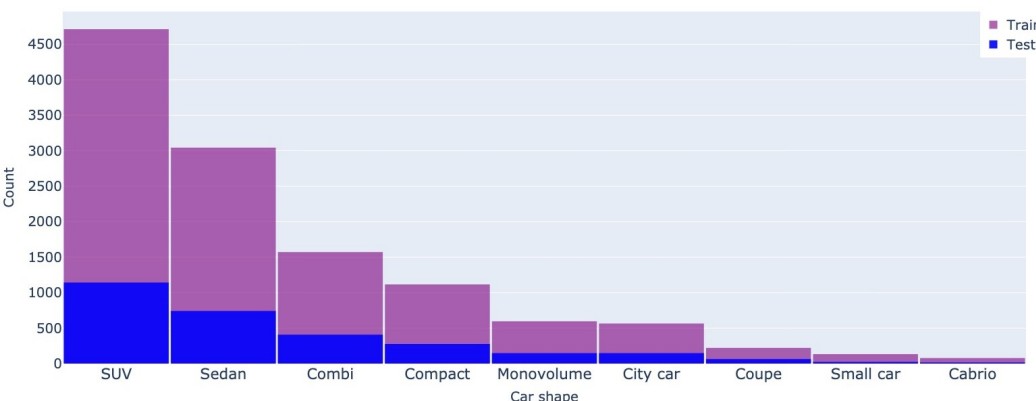

**Figure 9.** Car shape—Autovit.ro.

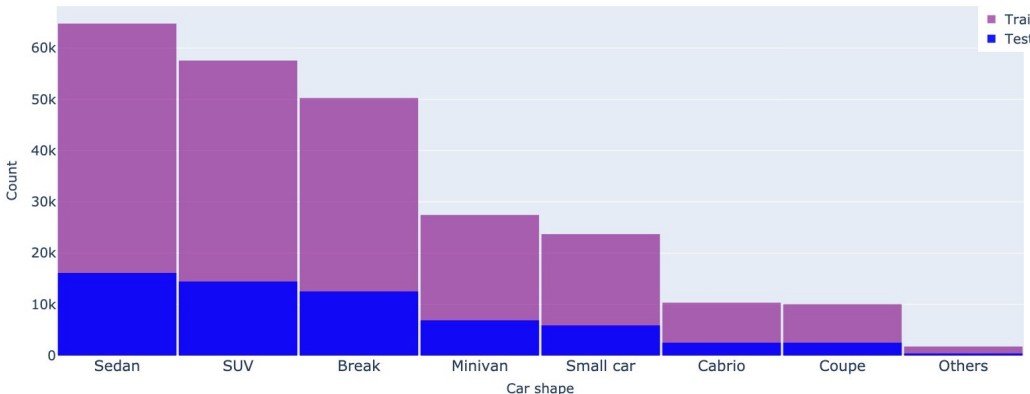

**Figure 10.** Car shape—Mobile.de.

The color did not have a high impact on the price, although specific car models had a default color with a lower price than other options (see Figures 11 and 12).

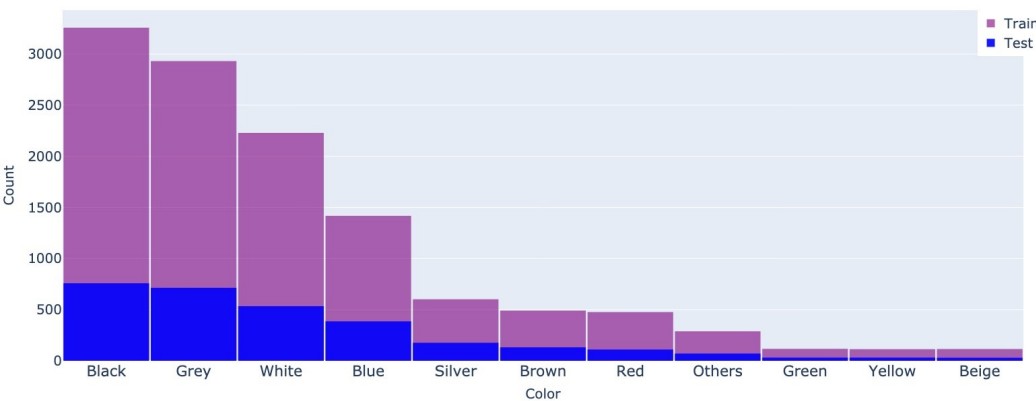

**Figure 11.** Color—Autovit.ro.

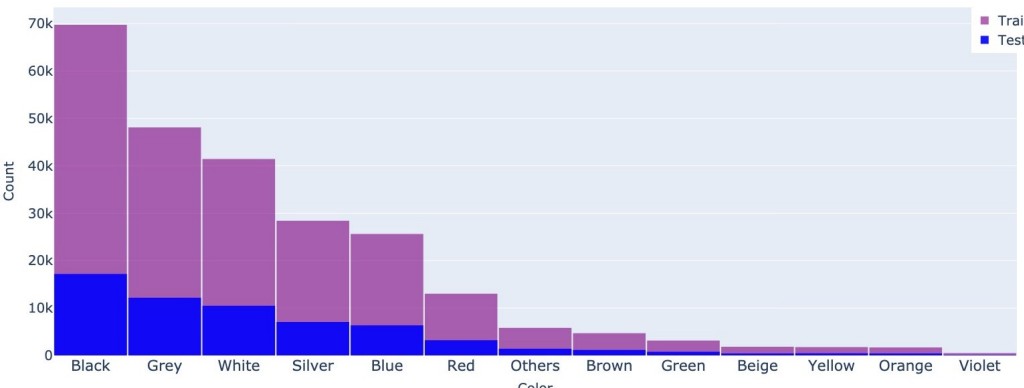

**Figure 12.** Color—Mobile.de.

In addition to the primary features, the vehicle owners may have appended a list of supplementary attributes for their cars in the advertisement. These add-ons were presented as an unordered list with string-based categorical values. The add-ons for a given vehicle could include up to 180 distinct categories for Autovit.ro and 120 for Mobile.de, with the most prevalent ones displayed in Table 2.

**Table 2.** Most frequent add-ons.

| Autovit.ro | | Mobile.de | |
| --- | --- | --- | --- |
| **Add-On** | **Count** | **Add-On** | **Count** |
| ABS | 13,626 | ABS | 300,662 |
| ESP | 13,494 | Power steering | 296,909 |
| Electric windows | 13,428 | Central locking | 295,256 |
| Radio | 12,933 | Electric windows | 293,416 |
| Driver airbag | 12,555 | Electric side mirror | 284,564 |
| Passenger airbag | 12,531 | ESP | 283,250 |
| Side airbag | 12,082 | Isofix | 263,770 |
| Heated exterior mirrors | 12,037 | On-board computer | 258,794 |
| Leather steering wheel | 11,986 | Alloy wheels | 248,468 |
| Isofix | 11,643 | Electric immobilizer | 247,690 |

Finally, the predicted feature was the price. Here, both datasets showcased similar distributions (see Figures 13 and 14), with the highest number of cars advertised at below EUR 20,000. It should be noted that the price represented the owner's asking price, so this may not have reflected the real market value of the car. Furthermore, the prices categorized by the most popular manufacturers displayed noteworthy variations in terms of value, with certain brands exhibiting a broad spectrum of potential prices. In contrast, other brands had values concentrated within a narrow range, as illustrated in Figure 15. It should also be noted that cars advertised on Autovit.ro had a higher mean asking price than cars of the same brand on Mobile.de. An underlying reason is that some of the cars from the Romanian market were bought from Germany and resold at a higher price in Romania.

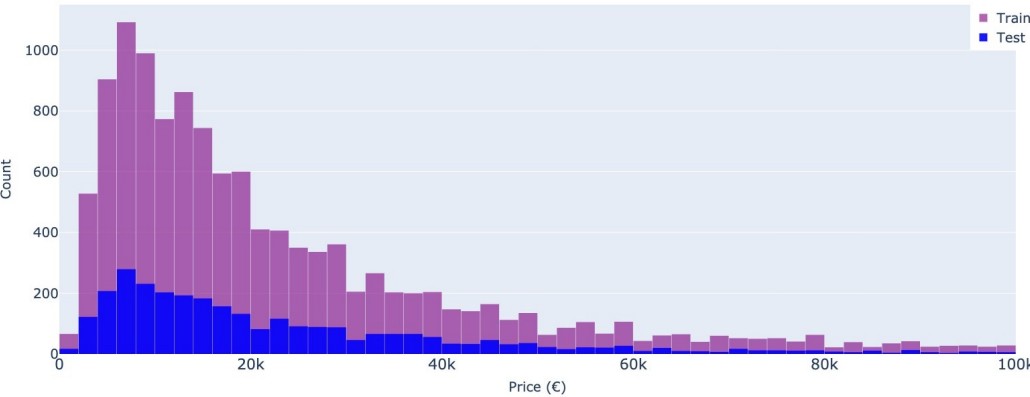

**Figure 13.** Price distribution—Autovit.ro.

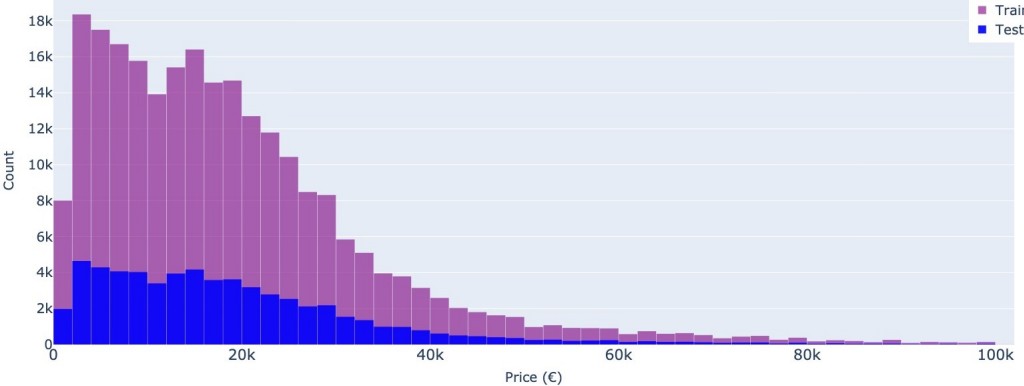

**Figure 14.** Price distribution—Mobile.de.

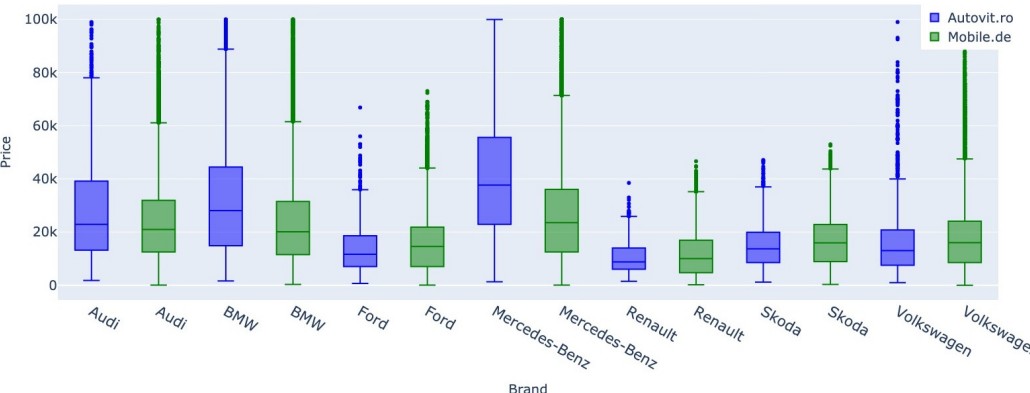

**Figure 15.** Price distribution—most popular brands.

Overall, we observed that the Mobile.de dataset had more evenly distributed numerical features and a more granular and diverse range of categorical features. In contrast, the Romanian market tended to be more biased towards certain car types.

The initial representation of the features remained largely consistent across all experiments, with only minor variations based on their respective types. Table 3 provides an overview of the features in their original state, while the subsequent model descriptions document any specific modifications made to them.

**Table 3.** Feature representation.

| Feature | Representation |
|---|---|
| Brand, model, gearbox, fuel, transmission, shape, color | Categorical features, represented with an integer as ID |
| Year of manufacture, mileage, engine power, engine capacity | Numerical features, scaled with Z-score in (0, 1) interval |
| Add-ons | List of categorical features, handled differently based on the approach |
| Images | .jpg file, converted into a 3D array |
| Price | Numerical feature, scaled differently based on the approach |

### 2.2. Gradient-Boosting Methods

The first experiments as a baseline for further analysis involved extreme gradient boosting. XGBoost [10] is an extension of the gradient-boosting method that combines multiple weak classifiers to form a strong classifier. The algorithm works by iteratively building decision trees based on the previous tree's error to minimize the model's overall error. Although it is widely used for classification, XGBoost can also be effectively used to predict continuous values in a regression task.

For the purpose of this experiment, we used all features described in Table 3, except the images, since XGBoost cannot handle this type of data efficiently. The IDs of the categorical features were scaled in the interval [0; 1] using a MinMaxScaler [31], since the algorithm did not have the ability to learn a better representation. The list of add-ons was hot-encoded as binary features to mark a specific extension's presence and account for different combinations of add-ons. The price was also scaled in the [0; 1] interval.

### 2.3. Neural Network Methods

We conducted various experiments involving different neural network architectures to enhance the learning of inter-feature relationships and optimize the representation of the car model and its manufacturer. All neural network architectures were trained to learn an embedding that optimally represented the car model and its brand. However, the training process failed to converge for certain infrequent car models. To address this issue,

we employed a mapping procedure whereby the embedding for models with fewer than 20 occurrences was learned for the manufacturer rather than for the model itself. To encode the name of the car more formally, we considered the following method:

$$name = \begin{cases} brand, & if\ freq(model) < 20 \\ model, & otherwise \end{cases} \tag{1}$$

Furthermore, the price was scaled based on the mean and standard deviation computed per car model (or brand, for models with a frequency of fewer than 20 entries). As such, the predicted price was computed as:

$$GT_i = \frac{price_i - mean_{name}}{std_{name}} \tag{2}$$

After the model performed the predictions, the final price was evaluated with the inverse transformation of Equation (2).

An embedding for the rest of the categorical features was learned during the training process. Numerical features were scaled as described in Table 3. In terms of add-ons, we experimented with several ways of including them in the training. The listing below offers a detailed description of how add-ons were encoded and used as features.

A general architecture (see Figure 16) was used in all the experiments. Unless a variation is mentioned regarding image processing or add-on representation, the neural network had the same general structure with different hyperparameters fine-tuned for each special case. The numerical features were used as-is, while an embedding layer trained its weights to learn an optimal representation for each categorical feature in the current context. The network component responsible for handling add-ons differed from one approach to another and is detailed for each variation. All these representations were concatenated and served as the input for a stack of dense layers of different sizes to learn and represent the complex relationships between the input features and their corresponding outputs. The final dense layer computed the model output and predicted the price value.

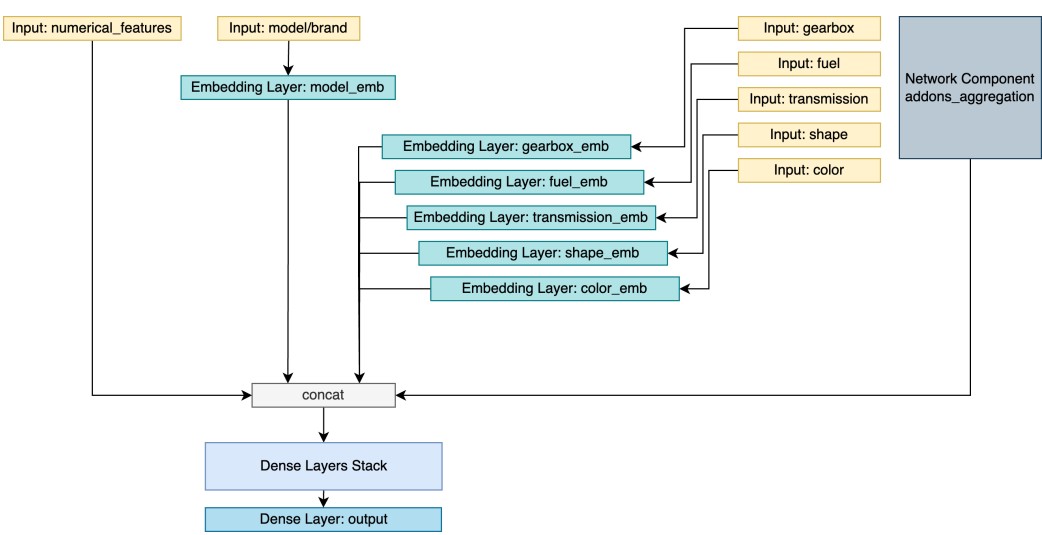

**Figure 16.** General architecture overview.

2.3.1. Neural Network with Hot-Encoded Add-On Projection

In this case (see Figure 17), the add-ons were hot-encoded to account for their presence or absence. We encoded them with $-1$ for their absence and 1 for their presence since these values were forwarded to a dense layer with a tanh activation function.

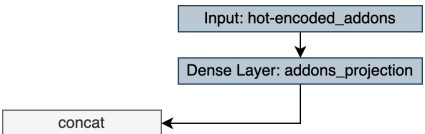

**Figure 17.** Hot-encoded add-on projection component.

### 2.3.2. Neural Network with Mean Add-On Learned Embeddings

In order to better learn a representation for each add-on and a representation of what the absence of that option meant, add-ons were aggregated using trainable embeddings (see Figure 18). For each potential add-on, two embeddings were computed: one to represent its presence and another to represent its absence. As a result, each vehicle was allocated an equal number of attributes pertaining to add-ons. The average of these embeddings was then considered an aggregation of the vehicle's characteristics.

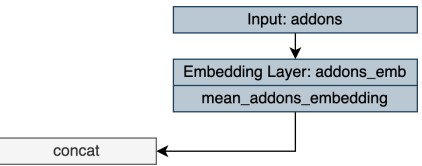

**Figure 18.** Mean add-on embedding component.

### 2.3.3. Neural Network with Add-On Embeddings and Multi-Head Self-Attention

Another approach similar to the previous method involved learning a contextualized representation of these add-ons (see Figure 19), which accounted for more relations than aggregating the average. As previously stated, a self-attention layer was used after computing the embeddings. The multi-head self-attention determined how much each individual add-on contributed to the representation of other add-ons, facilitating the identification of relevant dependencies and capturing long-range dependencies. Self-attention enabled the network to focus on different parts of the input adaptively. After the self-attention was applied, the output was averaged to obtain an aggregated representation.

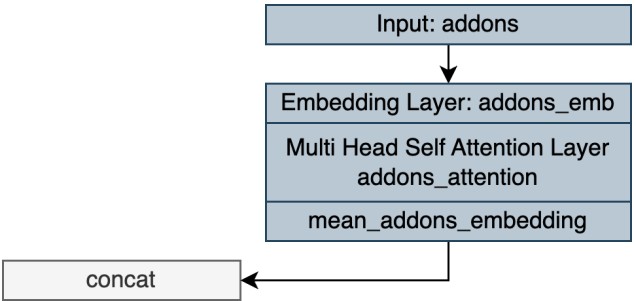

**Figure 19.** Add-on embedding with self-attention component.

### 2.3.4. Deep Neural Networks for Image Analysis and Add-On Multi-Head Self-Attention

In order to use all available information in the dataset, we adopted a comprehensive approach by incorporating numerical, categorical, and image attributes. The previously described features were aggregated using the above experimental method, the neural network with add-on embeddings and multi-head self-attention. Additionally, we integrated information about the vehicle's image into our model. To achieve this, the image was reshaped into a three-dimensional array of dimensions (224, 224, 3), enabling us to capture its visual characteristics. Subsequently, we employed a pre-trained convolutional neural network (CNN) architecture to extract contextualized information from the image. The resulting image projection was combined with the numerical projection obtained from the previous features. The concatenated attributes were then passed through a stack of dense layers, enabling the network to learn complex relationships and patterns within the data. Finally, the prediction was computed based on the processed features, resulting in a

comprehensive and informed output based on both the declared car characteristics and images provided in the ad.

For the purpose of this approach, we experimented with a pre-trained convolutional neural network architecture, namely EfficientNet [21], and a transformer-based architecture, Swin transformer [25]. Both architectures were selected based on their proven state-of-the-art results on various computer vision tasks and their transfer-learning ability. Although used for classification tasks, we removed the classification head from EfficientNet and used the last hidden states of both as a representation of the image characteristics. This representation was then sent into our downstream task of price regression. Although the weights of the pre-trained networks were initialized with their published values, we kept all layers trainable to allow the network to dynamically adjust and adapt to our specific task.

Figure 20 presents a detailed visual representation of this architecture.

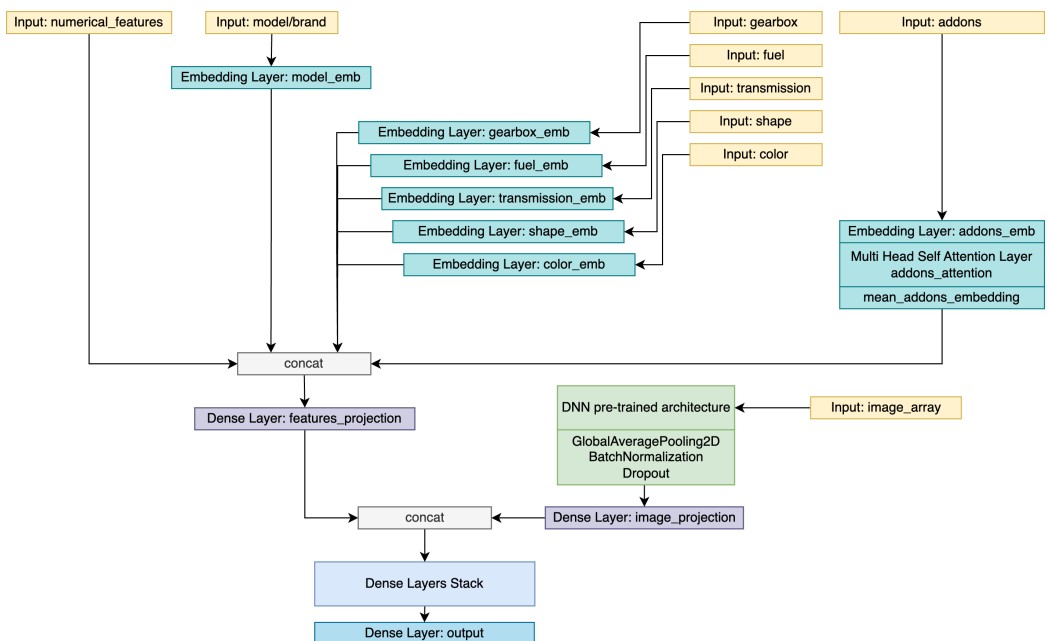

**Figure 20.** Deep neural networks for image and feature analysis architecture.

### 2.4. Experimental Setup

In all preceding models, we used the TensorFlow framework version 2.12 [32] to build the models and conducted hyperparameter tuning to achieve an optimal configuration by employing grid-search cross-validation via the Scikit-learn framework [31] and Keras Tuner [33]. Appendix A outlines the range of considered values and their corresponding optimal values.

### 3. Results

We computed the results (see Table 4) using the testing set for all described methods and their best hyperparameter settings. Three metrics were used to analyze the performance, namely the $R^2$ score to assess the prediction accuracy considering the variety of ground-truth labels, the mean absolute error (MAE), the median error (MedE), and the root mean squared error (RMSE). It should be noted that the results were calculated for the final prices with their ground truth after rescaling the models' predictions. Overall, the best-performing method on both datasets was the neural network with self-attention on add-on embeddings.

**Table 4.** Results (bold denotes the best model in terms of $R^2$ and MAE).

| Method | Autovit.ro | | | | Mobile.de | | | |
|---|---|---|---|---|---|---|---|---|
| | $R^2$ | MAE | MedE | RMSE | $R^2$ | MAE | MedE | RMSE |
| XGBoost | 0.92 | 2965 | 1613 | 5570 | 0.91 | 3102 | 1772 | 4527 |
| NN with add-on projection | 0.95 | 2463 | 1391 | 4403 | 0.94 | 2050 | 1236 | 3696 |
| NN with add-on embedding | 0.96 | 2234 | 1298 | 3788 | 0.95 | 2018 | 1220 | 3271 |
| NN with self-attention on add-on embedding | **0.96** | **2230** | **1296** | **3762** | **0.95** | **2012** | **1214** | **3261** |
| EfficientNet for image analysis and self-attention on add-on embedding | 0.96 | 2369 | 1350 | 3839 | 0.95 | 2030 | 1231 | 3306 |
| Swin transformer for image analysis and self-attention on add-on embedding | 0.96 | 2462 | 1533 | 3899 | 0.95 | 2116 | 1233 | 3340 |

## 4. Discussion

XGBoost represented our baseline against which to compare the results. The neural network architectures yielded the best results, as they learned more complex feature relations. The architecture that leveraged multi-head self-attention on add-on embeddings ranked the highest due to its ability to contextualize vehicle characteristics and gather information from different configurations. This approach achieved a high score since it was trained on multiple characteristics without overfitting.

The deep neural network that leveraged images along with numerical features also achieved high accuracy. However, as the features gathered from the images could be inferred from existing data, the last configuration did not improve upon the best-performing model prediction. A more detailed discussion on how the features impacted the model prediction, followed by an analysis of the best model's errors, as well as this study's limitations, are presented in the next subsections.

### 4.1. Ablation Study

As neural networks lack a straightforward method of determining feature importance, we performed an ablation study to highlight how each feature impacted the model and whether the neural network could learn to predict an accurate price without a particular piece of information.

Table 5 refers to the experiments carried out by removing a feature from the input while keeping the same model architecture of the best-performing approach described above. We kept the same brand and model of car in all cases, as these characteristics were compulsory for computing the final price. The most relevant features, whose absence impacted performance, were the year of manufacture, the mileage, and the list of add-ons. Other car characteristics did not disturb the predictions, as they could be inferred from a combination of the remaining features.

Moreover, we explored the extent to which each characteristic individually supported the predictions. Table 6 provides insights into how each feature influenced the price of a particular car model and brand. Again, the most influential features were the year of manufacture, mileage, and add-ons. However, all features influenced a car's price, more or less, and their importance was consistent across both datasets.

A particularly interesting experimental setup examined the model's ability to determine a car's price based on just car images. For this experiment, we considered the pre-trained deep neural networks and appended a dropout and a stack of fully connected layers (see Figure 21). The input for this network was the image array, and all the architecture weights were trainable. The results suggested that the model predicted the price of the vehicle with satisfactory accuracy using images alone. Moreover, the model attained a high

score for estimation accuracy in the case of the Mobile.de dataset, which contained over $50\times$ the number of image entries than the Autovit.ro dataset and was considerably more diverse.

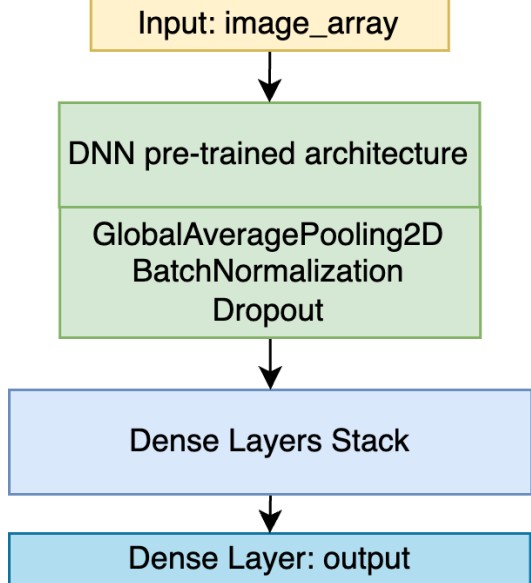

**Figure 21.** Deep neural network for image analysis architecture.

**Table 5.** Validation results after removing one feature (bold denotes features with the highest impact).

| Model Structure | Autovit.ro | | | | Mobile.de | | | |
|---|---|---|---|---|---|---|---|---|
| **Best architecture without:** | **$R^2$** | **MAE** | **MedE** | **RMSE** | **$R^2$** | **MAE** | **MedE** | **RMSE** |
| - | 0.96 | 2230 | 1296 | 3938 | 0.95 | 2012 | 1214 | 3399 |
| **Year of manufacture** | **0.94** | **3028** | **1949** | **4823** | **0.94** | **2223** | **1374** | **3833** |
| **Mileage** | **0.95** | **2723** | **1600** | **4403** | **0.93** | **2481** | **1589** | **4192** |
| Engine power | 0.96 | 2342 | 1354 | 4057 | 0.94 | 2077 | 1249 | 3698 |
| Engine capacity | 0.96 | 2251 | 1342 | 3958 | 0.95 | 2004 | 1203 | 3374 |
| **Add-ons** | **0.96** | **2401** | **1401** | **4164** | **0.93** | **2264** | **1370** | **3993** |
| Fuel | 0.96 | 2278 | 1336 | 3983 | 0.95 | 2007 | 1200 | 3382 |
| Transmission | 0.96 | 2243 | 1337 | 3947 | 0.95 | 2018 | 1221 | 3411 |
| Gearbox | 0.96 | 2270 | 1331 | 3974 | 0.94 | 2046 | 1248 | 3656 |
| Car shape | 0.96 | 2271 | 1315 | 3976 | 0.94 | 2051 | 1218 | 3671 |
| Color | 0.96 | 2299 | 1334 | 4002 | 0.95 | 2011 | 1202 | 3394 |

**Table 6.** Validation results after adding one feature (bold denotes features with the highest impact).

| Model Structure | Autovit.ro | | | | Mobile.de | | | |
|---|---|---|---|---|---|---|---|---|
| **Brand/Model with:** | **$R^2$** | **MAE** | **MedE** | **RMSE** | **$R^2$** | **MAE** | **MedE** | **RMSE** |
| - | 0.57 | 8761 | 5697 | 12914 | 0.57 | 7015 | 5182 | 9896 |
| **Year of manufacture** | **0.92** | **3444** | **1930** | **5570** | **0.87** | **3450** | **2240** | **5441** |
| **Mileage** | **0.88** | **4419** | **2872** | **6822** | **0.84** | **3955** | **2637** | **6036** |
| Engine power | 0.72 | 6525 | 3559 | 10421 | 0.75 | 4958 | 3226 | 7545 |
| Engine capacity | 0.63 | 7930 | 4947 | 11979 | 0.67 | 5828 | 3880 | 8669 |
| **Add-ons** | **0.82** | **5181** | **2927** | **8355** | **0.87** | **3374** | **2124** | **5441** |
| Fuel | 0.62 | 8010 | 5052 | 12140 | 0.61 | 6651 | 4852 | 9424 |
| Transmission | 0.60 | 8395 | 5215 | 12455 | 0.58 | 7083 | 5338 | 9780 |
| Gearbox | 0.61 | 8113 | 4725 | 12298 | 0.63 | 6427 | 4583 | 9179 |
| Car shape | 0.59 | 8546 | 5476 | 12610 | 0.60 | 6804 | 4945 | 9544 |
| Color | 0.59 | 8659 | 5669 | 12706 | 0.59 | 6784 | 4873 | 9663 |
| **Images (EfficientNet)** | **0.69** | **6899** | **4053** | **10965** | **0.82** | **4045** | **2658** | **6402** |
| **Images (Swin transformer)** | **0.67** | **7534** | **4335** | **11313** | **0.82** | **4087** | **2697** | **6471** |

### 4.2. Transfer-Learning Capabilities

The evaluation results considering only car images from Autovit.ro were unsatisfactory and differed from the high scores computed for the Mobile.de dataset. As such, we tested whether the training carried out on the larger dataset had transfer-learning capabilities and could improve the results for the other market. In this regard, we took the architecture from Figure 21 and pre-trained it on the Mobile.de images to predict the price. After this, we fine-tuned the resulting model on the Autovit.ro dataset for only five epochs to gather the particularities of the Romanian car prices and market. The results presented in Table 7 showcased an improvement of over 10% in the $R^2$ score for the Autovit.ro validation set after fine-tuning on the Mobile.de pre-trained architecture. This suggested that our method led to increased performance and adaptation capabilities for low-resource datasets. Therefore, our method was particularly useful for underdeveloped car markets lacking diversity and coverage.

**Table 7.** Validation results for transfer learning (bold denotes the best performance).

| Method | Autovit.ro | | | |
|---|---|---|---|---|
| | $R^2$ | MAE | MedE | RMSE |
| DNN for image analysis without pre-training | 0.69 | 6899 | 4053 | 10965 |
| DNN for image analysis with pre-training and fine-tuning | **0.79** | **5652** | **3373** | **9024** |

### 4.3. Error Analysis

Quantile–quantile plots (see Figures 22 and 23) were used to analyze the prediction values in relation to the ground truth; the majority of price estimations were gathered along the main diagonal for both datasets. This indicated a robust prediction mechanism, even for highly varied data.

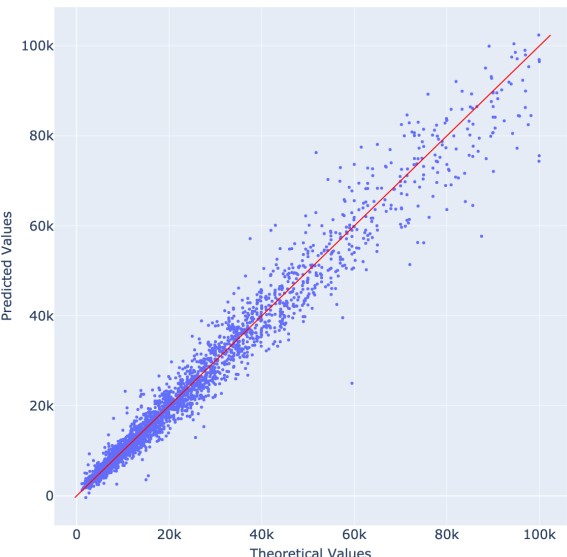

**Figure 22.** Q–Q plot—Autovit.ro.

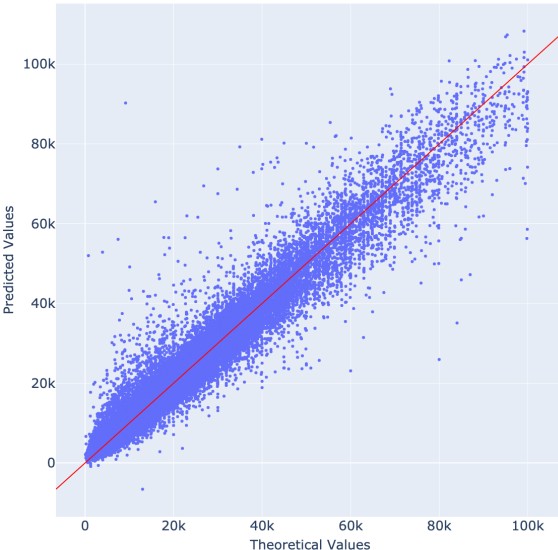

**Figure 23.** Q–Q plot—Mobile.de.

Figures 24 and 25 highlight the mean absolute error per brand. As expected, the highest errors were among luxury cars with a high mean price; consequently, the prediction error was relative to the price. For both datasets, brands such as Bentley, Aston Martin, Ferrari, and Porche were among those that increased the mean error. Moreover, under-represented brands in the dataset, such as Alpina and GMC, also yielded higher than usual errors due to the lack of training examples required for contextualizing information about them. Nevertheless, we observed that besides these particular cases of luxury or under-represented cars, the models' errors fell below the average, and their estimation range was satisfactory.

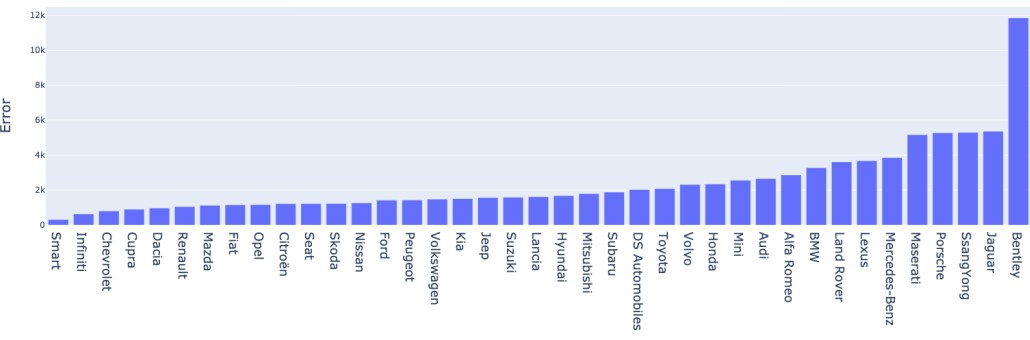

**Figure 24.** Mean absolute error by brand—Autovit.ro.

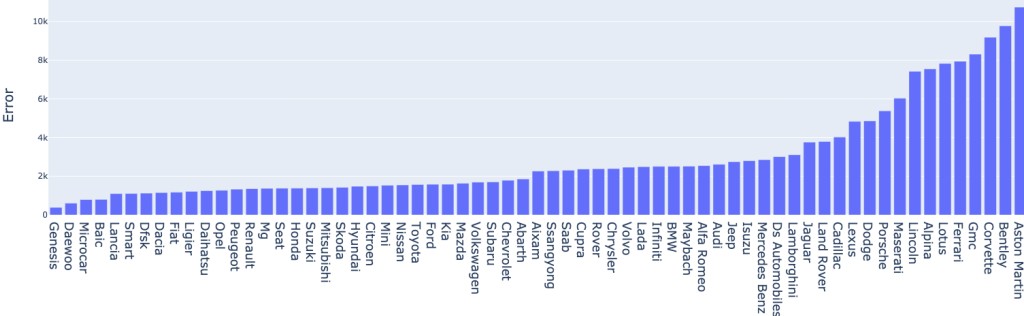

**Figure 25.** Mean absolute error by brand—Mobile.de.

*4.4. Limitations*

The limitations of the model's performance were strongly connected to and derived from the datasets' shortcomings. Both datasets were scraped from used car advertisement websites, so they were inherently prone to flaws in human judgment and the individual assessment of each vehicle's value. The seller could have overestimated the price of the car or aimed for an expedited transaction with a lower price; thus, the ground truth for prediction may not have been the real market value of the vehicle. Moreover, the users may not have specified the hidden flaws of the vehicle, which may have also affected the estimation. Another detrimental aspect was that some car ads had an incomplete or non-existent list of add-ons. As there were over 100 possible add-ons for each website to choose from, this cumbersome task was often overlooked by sellers; this resulted in ads containing declared features that were not correlated with the price. As Table 5 indicates, the add-ons had a big impact on the price predictions, and the lack of an accurate list of these add-ons would degrade the models' performance. However, the studied architectures performed well on the two datasets, highlighting their ability to leverage complex features and their potential for learning using different datasets. Moreover, the scores did not improve as we experimented with larger models and more fine-grained architectures; this indicated the limited correctness of our datasets. Since the ads were scraped on March 2023, the information represented the state of the ads at that time and did not account for purchases or changes made to the ads after that date.

A limitation of the considered deep neural models was their marginal decrease in performance when adding images. More research is required to learn how to best represent the visual features of car sale ads and search for hidden flaws or aspects that do not appear in the numerical entries.

## 5. Conclusions and Future Work

This research successfully reached its objective of developing comprehensive and accurate deep learning models for predicting used car prices in different car markets. We built upon our initial study [8] in terms of the considered datasets, feature aggregation methods, and image analysis. The construction of the largest datasets to date, comprising over 300,000 entries from the German market and 15,000 entries from the Romaian market, provided valuable resources for training deep prediction models and enabled a comparative analysis between these two markets. Our approaches achieved a high prediction accuracy, with an $R^2$ score exceeding 0.95. Incorporating multiple features in these models contributed to their effectiveness and reliability in accurately estimating used car prices. Furthermore, we showcased the potential of using convolutional architectures to predict car prices based solely on visual features extracted from car images. Notably, this model exhibited transfer-learning capabilities, leading to an improved prediction accuracy, particularly in cases where the training datasets had limited resources. The findings emphasized the importance of visual information in accurately predicting car prices and have practical implications for buyers and sellers in assessing the value of vehicles.

Several paths for future research can be pursued to build upon the findings and contributions of this study. First, a method for dataset standardization across different car markets and features should be developed. Given the variations in data collection practices and feature representation across different markets, establishing standardized data pre-processing and feature engineering protocols would enhance comparability and enable more robust analyses. As such, developing a standardized approach would increase the generalizability of the prediction models. Second, future research should aim to incorporate economic situations and inflation rates into the prediction models. Integrating economic indicators into the models would enable a more comprehensive understanding of the pricing dynamics. Furthermore, an ablation study focusing on the visual features extracted from car images should be conducted to determine the relative importance of different image parts in price prediction. This would provide valuable insights into the decision-making processes of buyers and sellers. Moreover, we envision future research avenues to

extract the car manufacturer and model directly from the picture and improve prediction quality from images only.

**Author Contributions:** Conceptualization, A.D., A.C., S.R. and M.D.; data curation, A.D. and A.C.; formal analysis, A.D. and A.C.; funding acquisition, M.D.; investigation A.D.; methodology, A.D.; project administration, M.D.; resources, S.R. and M.D.; software, A.D.; supervision, M.D.; validation, S.R., L.-M.N. and M.D.; visualization, A.D.; writing—original draft, A.D., A.C. and D.I.; writing—review and editing, S.R., L.-M.N., V.G. and M.D. All authors have read and agreed to the published version of the manuscript.

**Funding:** This work was funded by the "Automated car damage detection and cost prediction—InsureAI"/"Detectia automata a daunelor si predictia contravalorii aferente–InsureAI" project, contract number 30/221_ap3/22.07.2022, MySMIS code: 142909.

**Institutional Review Board Statement:** Not applicable.

**Informed Consent Statement:** Not applicable.

**Data Availability Statement:** The data presented in this study are available on request from the corresponding author.

**Conflicts of Interest:** The authors declare no conflict of interest.

## Abbreviations

The following abbreviations are used in this manuscript:

| | |
|---|---|
| BPNN | Backpropagation neural network |
| CNN | Convolutional neural network |
| DNN | Deep neural network |
| GRA | Grey relation analysis |
| GT | Ground truth |
| HOG | Histogram of oriented gradients |
| HP | Horsepower |
| KNN | K nearest neighbor |
| MAE | Mean absolute error |
| MedE | Median error |
| NN | Neural network |
| PSO | Particle swarm optimization |
| RMSE | Root mean squared error |
| SVM | Support vector machines |

## Appendix A

**Table A1.** Hyperparameter search options.

| Method | Hyperparameters | |
|---|---|---|
| | **Autovit.ro** | **Mobile.de** |
| XGBoost | learning rate: 0.12 (0.10–0.15) <br> max depth: 14 (12–16) <br> min child weight: 8 (6–9) <br> subsample: 1 (0.7–1) <br> colsample bytree: 0.8 (0.6–1) <br> objective: squared error | learning rate: 0.12 (0.10–0.15) <br> max depth: 16 (12–18) <br> min child weight: 8 (6–9) <br> subsample: 1 (0.7–1) <br> colsample bytree: 0.9 (0.6–1) <br> objective: squared error |
| NN with add-on projection | learning rate: $10^{-3}$ ($10^{-2}$–$10^{-4}$) <br> add-on projection: $2^5$ ($2^4$–$2^8$) <br> model embedding: $2^6$ ($2^4$–$2^7$) <br> fuel embedding: $2^1$ ($2^1$–$2^3$) <br> transmission embedding: $2^1$ ($2^1$–$2^3$) <br> gearbox embedding: $2^2$ ($2^1$–$2^3$) <br> car shape embedding: $2^1$ ($2^1$–$2^3$) <br> color embedding: $2^2$ ($2^1$–$2^3$) <br> dense layer 1: $2^8$ ($2^7$–$2^9$) <br> dense layer 2: $2^7$ ($2^6$–$2^8$) <br> dense layer 3: $2^4$ ($2^3$–$2^5$) | learning rate: $10^{-3}$ ($10^{-2}$–$10^{-4}$) <br> add-on projection: $2^6$ ($2^4$–$2^8$) <br> model embedding: $2^6$ ($2^4$–$2^7$) <br> fuel embedding: $2^2$ ($2^1$–$2^3$) <br> transmission embedding: $2^2$ ($2^1$–$2^3$) <br> gearbox embedding: $2^1$ ($2^1$–$2^3$) <br> car shape embedding: $2^3$ ($2^1$–$2^3$) <br> color embedding: $2^2$ ($2^1$–$2^3$) <br> dense layer 1: $2^9$ ($2^7$–$2^9$) <br> dense layer 2: $2^7$ ($2^6$–$2^8$) <br> dense layer 3: $2^4$ ($2^3$–$2^5$) |

**Table A1.** *Cont.*

| Method | Hyperparameters | |
|---|---|---|
| | **Autovit.ro** | **Mobile.de** |
| NN with add-on embedding | learning rate: $10^{-2}$ ($10^{-2}$–$10^{-4}$)<br>add-on embedding: $2^5$ ($2^4$–$2^8$)<br>model embedding: $2^4$ ($2^4$–$2^7$)<br>fuel embedding: $2^1$ ($2^1$–$2^3$)<br>transmission embedding: $2^1$ ($2^1$–$2^3$)<br>gearbox embedding: $2^1$ ($2^1$–$2^3$)<br>car shape embedding: $2^2$ ($2^1$–$2^3$)<br>color embedding: $2^2$ ($2^1$–$2^3$)<br>dense layer 1: $2^9$ ($2^7$–$2^9$)<br>dense layer 2: $2^6$ ($2^6$–$2^8$)<br>dense layer 3: $2^4$ ($2^3$–$2^5$) | learning rate: $10^{-2}$ ($10^{-2}$–$10^{-4}$)<br>add-on embedding: $2^5$ ($2^4$–$2^8$)<br>model embedding: $2^4$ ($2^4$–$2^7$)<br>fuel embedding: $2^1$ ($2^1$–$2^3$)<br>transmission embedding: $2^1$ ($2^1$–$2^3$)<br>gearbox embedding: $2^1$ ($2^1$–$2^3$)<br>car shape embedding: $2^2$ ($2^1$–$2^3$)<br>color embedding: $2^2$ ($2^1$–$2^3$)<br>dense layer 1: $2^9$ ($2^7$–$2^9$)<br>dense layer 2: $2^6$ ($2^6$–$2^8$)<br>dense layer 3: $2^4$ ($2^3$–$2^5$) |
| NN with self-attention on add-on embedding | learning rate: $10^{-2}$ ($10^{-2}$–$10^{-4}$)<br>add-on embedding: $2^5$ ($2^4$–$2^8$)<br>add-on attention heads: $2^1$ ($2^1$–$2^3$)<br>add-on attention key: $2^3$ ($2^2$–$2^5$)<br>model embedding: $2^4$ ($2^4$–$2^7$)<br>fuel embedding: $2^1$ ($2^1$–$2^3$)<br>transmission embedding: $2^1$ ($2^1$–$2^3$)<br>gearbox embedding: $2^1$ ($2^1$–$2^3$)<br>car shape embedding: $2^2$ ($2^1$–$2^3$)<br>color embedding: $2^2$ ($2^1$–$2^3$)<br>dense layer 1: $2^9$ ($2^7$–$2^9$)<br>dense layer 2: $2^6$ ($2^6$–$2^8$)<br>dense layer 3: $2^4$ ($2^3$–$2^5$) | learning rate: $10^{-2}$ ($10^{-2}$–$10^{-4}$)<br>add-on embedding: $2^6$ ($2^4$–$2^8$)<br>add-on attention heads: $2^1$ ($2^1$–$2^3$)<br>add-on attention key: $2^5$ ($2^2$–$2^6$)<br>model embedding: $2^5$ ($2^4$–$2^7$)<br>fuel embedding: $2^2$ ($2^1$–$2^3$)<br>transmission embedding: $2^2$ ($2^1$–$2^3$)<br>gearbox embedding: $2^1$ ($2^1$–$2^3$)<br>car shape embedding: $2^3$ ($2^1$–$2^3$)<br>color embedding: $2^2$ ($2^1$–$2^3$)<br>dense layer 1: $2^9$ ($2^7$–$2^9$)<br>dense layer 2: $2^7$ ($2^6$–$2^8$)<br>dense layer 3: $2^3$ ($2^3$–$2^5$) |
| DNN for image analysis and self-attention on add-on embedding | learning rate: $10^{-3}$ ($10^{-2}$–$10^{-4}$)<br>feature projection: $2^9$ ($2^8$–$2^{10}$)<br>image projection: $2^9$ ($2^8$–$2^{10}$)<br>dense layer 1: $2^9$ ($2^7$–$2^9$)<br>dense layer 2: $2^8$ ($2^6$–$2^8$)<br>dense layer 3: $2^4$ ($2^3$–$2^5$) | learning rate: $10^{-3}$ ($10^{-2}$–$10^{-4}$)<br>feature projection: $2^9$ ($2^8$–$2^{10}$)<br>image projection: $2^9$ ($2^8$–$2^{10}$)<br>dense layer 1: $2^9$ ($2^7$–$2^9$)<br>dense layer 2: $2^8$ ($2^6$–$2^8$)<br>dense layer 3: $2^4$ ($2^3$–$2^5$) |
| DNN for image analysis and self-attention on add-on embedding | learning rate: $10^{-5}$ ($10^{-3}$–$10^{-6}$)<br>dense layer 1: $2^{10}$ ($2^7$–$2^{10}$)<br>dense layer 2: $2^9$ ($2^7$–$2^{10}$)<br>dense layer 3: $2^6$ ($2^5$–$2^8$)<br>dense layer 4: $2^4$ ($2^3$–$2^5$) | learning rate: $10^{-5}$ ($10^{-3}$–$10^{-6}$)<br>dense layer 1: $2^{10}$ ($2^7$–$2^{10}$)<br>dense layer 2: $2^9$ ($2^7$–$2^{10}$)<br>dense layer 3: $2^6$ ($2^5$–$2^8$)<br>dense layer 4: $2^4$ ($2^3$–$2^5$) |

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
