# Peer review of "Car Price Quotes Driven by Data-Comprehensive Predictions Grounded in Deep Learning Techniques"

_electronics, doi:10.3390/electronics12143083_

Round 1

Reviewer 1 Report

Thank you for your submission. This is an interesting paper.

-          However, it is not objective enough and the method is vague. You have used all the jargons of AI algorithms from PSO to GBoosting; there is no need to apply everything to have a proper publication. You would need to relook at the way you have presented the paper and properly structure it to highlight what it is you are doing.

-          You should have tables to report the test errors for the methods.

-          What software is used for analysis? Please mention.

-          It seems that the paper is written by people who are in business or finance and have no idea what AI methods are and are aimed and designed to do, as an example, Back propagation NNs and CNNs, CNNs are a type of BPNNs. Every NN is trained based on a type of BP algorithm. Please use these references to see how an applied AI paper is written and structures.

-          Use these recent references to improve your background on applications of AI methods:

Shafiabady, N., Hadjinicolaou, N., Ud Din, F., Bhandari, B., Wu, R. M. X., & Vakilian, J. (2023). Using Artificial Intelligence (AI) to predict organizational agility. PLoS One, 18(5 May), e0283066. [e0283066]. https://doi.org/10.1371/journal.pone.0283066

Wu, R. M. X., Wang, Y., Shafiabady, N., Zhang, H., Yan, W., Gou, J., Shi, Y., Liu, B., Gide, E., Kang, C., Zhang, Z., Shen, B., Li, X., Fan, J., He, X., Soar, J., Zhao, H., Sun, L., Huo, W., & Wang, Y. (2023). Using multi-focus group method as an effective tool for eliciting business system requirements: Verified by a case study. PLoS One, 18 (3 March), [e0281603]. https://doi.org/10.1371/journal.pone.0281603

Bhandari, B., Park, G., & Shafiabady, N. (2023). Implementation of transformer-based deep learning architecture for the development of surface roughness classifier using sound and cutting force signals. Neural Computing and Applications, 35(18), 13275-13292. https://doi.org/10.1007/s00521-023-08425-z

Zaidi, S. F. M., Shafiabady, N., & Beilby, J. (2023). Identifying presence of cybersickness symptoms using AI-based predictive learning algorithms. Virtual Reality, 1-8. https://doi.org/10.1007/s10055-023-00813-z

-          Please submit a plain copy and a word copy with track changes on so that the reviewer can identify the changes you have made upon implementing the changes.

-          The methods in the paper need to be reinvestigated and the results be presented professionally so that it makes sense to an AI expert in addition to general audience.

Thank you.

English wise, the paper is OK.

Author Response

Thank you for your submission. This is an interesting paper.
Authors’ Response: Thank you kindly.

-          However, it is not objective enough and the method is vague. You have used all the jargons of AI algorithms from PSO to GBoosting; there is no need to apply everything to have a proper publication. You would need to relook at the way you have presented the paper and properly structure it to highlight what it is you are doing.
Authors’ Response: We thank the reviewer for pointing out the lack of clarity regarding the methods we use in our study. We have added additional clarifications in the Introduction section (lines 234-238) highlighting the fact that we only experiment with Gradient Boosting as baseline, Neural Networks for learning inter-feature relationships, and Deep Neural Networks for extracting relevant visual features from the images. While we provide an extensive ablation study on these 3 approaches, we disagree with the comment that we “apply everything”. The other aspects the reviewer is referring to, such as PSO, are not employed in our evaluation, but only described in the Related Work section. We presented an extensive literature review of a wide variety of methods that have been employed in the past for car price prediction. It seems that the aspects you are referring to, for eg. PSO, are concepts that we presented in the literature review, not on our proposed methods. We have studied and experimented with three methods, namely Gradient Boosting, Neural Networks and Deep Neural Networks for image analysis.

-          You should have tables to report the test errors for the methods.
Authors’ Response: For our proposed methods, Tables 4-7 describe the performance on the test set in terms of R2, Mean Absolute Error, and Median Error.

-          What software is used for analysis? Please mention.
Authors’ Response:  We did not use any dedicated software for the analysis, we performed the computations using Python programming language, and several frameworks stated in the Method and Result chapters - such as Tensorflow, Scikit-learn or Keras Tuner.

-          It seems that the paper is written by people who are in business or finance and have no idea what AI methods are and are aimed and designed to do, as an example, Back propagation NNs and CNNs, CNNs are a type of BPNNs. Every NN is trained based on a type of BP algorithm. Please use these references to see how an applied AI paper is written and structures.
Authors’ Response: We have revised the presentation of the method. We only used the BP (Back Propagation) acronym in the Related Work section when talking about the work of Liu et al. [5] in order to align with the terminology of the authors.  On a side note, we are computer scientists with AI as the core of our research ?

-          Use these recent references to improve your background on applications of AI methods:
Shafiabady, N., Hadjinicolaou, N., Ud Din, F., Bhandari, B., Wu, R. M. X., & Vakilian, J. (2023). Using Artificial Intelligence (AI) to predict organizational agility. PLoS One, 18(5 May), e0283066. [e0283066]. https://doi.org/10.1371/journal.pone.0283066
Wu, R. M. X., Wang, Y., Shafiabady, N., Zhang, H., Yan, W., Gou, J., Shi, Y., Liu, B., Gide, E., Kang, C., Zhang, Z., Shen, B., Li, X., Fan, J., He, X., Soar, J., Zhao, H., Sun, L., Huo, W., & Wang, Y. (2023). Using multi-focus group method as an effective tool for eliciting business system requirements: Verified by a case study. PLoS One, 18 (3 March), [e0281603]. https://doi.org/10.1371/journal.pone.0281603
Bhandari, B., Park, G., & Shafiabady, N. (2023). Implementation of transformer-based deep learning architecture for the development of surface roughness classifier using sound and cutting force signals. Neural Computing and Applications, 35(18), 13275-13292. https://doi.org/10.1007/s00521-023-08425-z
Zaidi, S. F. M., Shafiabady, N., & Beilby, J. (2023). Identifying presence of cybersickness symptoms using AI-based predictive learning algorithms. Virtual Reality, 1-8. https://doi.org/10.1007/s10055-023-00813-z
Authors’ Response: We read the previous studies, thank you for your suggestion.

-          Please submit a plain copy and a word copy with track changes on so that the reviewer can identify the changes you have made upon implementing the changes.
Authors’ Response: We have generated a PDF with track changes between the 2 versions.

-          The methods in the paper need to be reinvestigated and the results be presented professionally so that it makes sense to an AI expert in addition to general audience.
Authors’ Response: We tried our best to emphasize our findings. If you have any additional specific suggestions, please let us know, and we are more than happy to process them.

Thank you.

Reviewer 2 Report

Title:  Car Price Quotes Driven by Data – Comprehensive Predictions Grounded in Deep Learning Techniques

 ID: electronics-2454454

 In this article, the authors developed a machine learning models for predicting the prices of used cars in different markets. They used two data sets from Romania and Germany, and proposed ways of aggregating additional features compared with previous studies. The authors also introduced the state-of-the-art approaches for developing prediction models. Further, the created a baseline model to predict the car prices based on visual features extracted from car image.   The authors considered several methods to predict the car prices and used three metrics to evaluate the performance of each method and found that the neural network with self-attention on adds-ons embedding performed the best on both data sets.

 The manuscript is a good contribution to literature, and both sellers and buyers can benefit from this research. However, below are some comments that are expected to improve the manuscript.

        I.            General Comments

a)      When extracting the data from car image-ads, can the authors grantee cars images exist in the ad are the latest ones? Or maybe old? Is there any to check this issue?

b)      Page 4: Last paragraph: Line 178: it seems that Dutulescu et al[8] had previous study about the car price prediction using also Romanian website regarding used cars? Is it possible to compare apply your method to that data and compare the performance of the best one they got with the best method you got in this manuscript?

c)      Is it possible to have more than one ad for the same car in any of the websites you used? Or not allowed? Or the algorithm used to extract the data can detect this issue?

d)      I noticed R2 is used as performance measure, it is recommended to add the adjusted R2 for each model, then compare different models, as this value will not increase unless the variable is really contributing to the value of R2.

       I.            Specific comments:

a)      On page 3: Lines 129,134, 135: define the RMSE and MAE before using it, then you can use it later in page 4 Line 170.

b)      On page 4: Line 141: change the word “bard” to brand.

c)      On page 7: Line 313: … the range is defined as mean model+- Std Model, the terms of this equation should be explained, the mean and standard deviation of what? Is it of  the model year? Or model price? Or something else?

d)      On pages: 10-12: all Figures 7-12: it is better to show bar charts in these figures in descending order(from the one with the highest frequency (count) to the one with the lowest frequency (count)). This will make it easy for the reader to discover the effect of variable(car shape, and the colour).

e)      On page 16:Line 433: “ Instead of a 0-1 encoding, we opted to encode them with -1 for their absence and 1 for their presence”. The codes should be 1 and 0, and not 1 for both cases.

f)       On page 17: Line 477: remove the space at the beginning of this line.

No comments about the English language, everything is okay. 

Author Response

In this article, the authors developed a machine learning models for predicting the prices of used cars in different markets. They used two data sets from Romania and Germany, and proposed ways of aggregating additional features compared with previous studies. The authors also introduced the state-of-the-art approaches for developing prediction models. Further, the created a baseline model to predict the car prices based on visual features extracted from car image.   The authors considered several methods to predict the car prices and used three metrics to evaluate the performance of each method and found that the neural network with self-attention on adds-ons embedding performed the best on both data sets. 
 The manuscript is a good contribution to literature, and both sellers and buyers can benefit from this research. However, below are some comments that are expected to improve the manuscript.
Authors’ Response: Thank you kindly for your appreciation.

        I.            General Comments 

a)      When extracting the data from car image-ads, can the authors grantee cars images exist in the ad are the latest ones? Or maybe old? Is there any to check this issue?
Authors’ Response: The car images are the ones existing for the ad at the time of the scraping. Sellers are able to update their images, and they tend to be the most recent image of the vehicle. We stated in the Limitations section that the information was the most recent update at the time of the scraping, March 2023.

b)      Page 4: Last paragraph: Line 178: it seems that Dutulescu et al[8] had previous study about the car price prediction using also Romanian website regarding used cars? Is it possible to compare apply your method to that data and compare the performance of the best one they got with the best method you got in this manuscript?
Authors’ Response: The dataset presented by Dutulescu et al. was scraped long before March 2023, so there are differences in existing car ads. Moreover, our initial version of the dataset lacked some of the relevant features that we take into consideration (e.g. engine power), which were subsequently extracted. As such, we cannot run the new models on the initial dataset and we cannot fill-in the missing details since the ads are no longer available.

c)      Is it possible to have more than one ad for the same car in any of the websites you used? Or not allowed? Or the algorithm used to extract the data can detect this issue?
Authors’ Response: As per the Terms and Conditions stated by each website, posting duplicate ads of the same vehicle is not allowed, and there are methods in place to remove those ads. Based on this assumption, we conclude that no duplicate ads are part of our dataset. We have added this statement in the paper, in Section 2.1 where we describe our dataset.

d)      I noticed R2 is used as performance measure, it is recommended to add the adjusted R2 for each model, then compare different models, as this value will not increase unless the variable is really contributing to the value of R2. 
Authors’ Response: After extensive consideration, we concluded that the adjusted R2 could not be computed in this context. The models we employ are non-linear deep neural networks with large numbers of trainable weights. After an extensive search, we could not find a way to determine the degrees of freedom (a component of the adjusted R2 formula) in these networks. Our approaches do not involve linear models to calculate this value. However, the soundness of our approach and the correctness of the R2 value meaning is backed by the fact that we report the results on a separate test partition (Section 2.1 explains how the partition was made) that does not contain ads existing in the train partition. (https://www.statsmodels.org/stable/generated/statsmodels.regression.linear_model.RegressionResults.rsquared_adj.html#statsmodels.regression.linear_model.RegressionResults.rsquared_adj,  https://stats.stackexchange.com/questions/57027/what-does-degree-of-freedom-mean-in-neural-networks )

       I.            Specific comments: 

a)      On page 3: Lines 129,134, 135: define the RMSE and MAE before using it, then you can use it later in page 4 Line 170. DONE
b)      On page 4: Line 141: change the word “bard” to brand. DONE
c)      On page 7: Line 313: … the range is defined as mean model+- Std Model, the terms of this equation should be explained, the mean and standard deviation of what? Is it of the model year? Or model price? Or something else? It is the mean and std of the model’s price. Modified in the PDF also.
d)      On pages: 10-12: all Figures 7-12: it is better to show bar charts in these figures in descending order(from the one with the highest frequency (count) to the one with the lowest frequency (count)). This will make it easy for the reader to discover the effect of variable(car shape, and the colour). Figures 7 and 8 are histograms, not bar charts, so the order of the bins should be in the ascending order of the x-values. We have changed the Figures 9 - 12 (the car shape and the color) according to your suggestion.
e)      On page 16:Line 433: “ Instead of a 0-1 encoding, we opted to encode them with -1 for their absence and 1 for their presence”. The codes should be 1 and 0, and not 1 for both cases. We have stated more clearly our approach.
f)       On page 17: Line 477: remove the space at the beginning of this line. We could not find this problem.

Authors’ Response: All previous changes were made. 

Round 2

Reviewer 1 Report

The previous comments I had mentioned haven't been addressed.

New comment: RMSE has been metioned but the tables are showing MAE only. Is there a frame of reference for comparison with other works?

The English is fine. The paper needs just one more round of proof reading.

Author Response

The previous comments I had mentioned haven't been addressed.
Response: We tried our best to address all due changes and argued each modification accordingly. We have provided an extensive response to each suggestion. Please let us know specifically what has not been addressed and how we may improve further our manuscript.

New comment: RMSE has been metioned but the tables are showing MAE only. Is there a frame of reference for comparison with other works?
Response: RMSE was mentioned in the related works subsection. We considered it redundant with the other reported metrics for the new experiments. Nevertheless, we agree that having it for future comparisons is better. As such, we have added RMSE values in all result tables.